# Adapted Fringe Projection Sequences for Changing Illumination Conditions on the Example of Measuring a Wrought-Hot Object Influenced by Forced Cooling

**DOI:** 10.3390/s21051599

**Published:** 2021-02-25

**Authors:** Lorenz Quentin, Rüdiger Beermann, Carl Reinke, Pascal Kern, Markus Kästner, Eduard Reithmeier

**Affiliations:** Institute of Measurement and Automatic Control, Leibniz Universität Hannover, Nienburger Str. 17, D-30167 Hannover, Germany; beermann@imr.uni-hannover.de (R.B.); Carl.Reinke@gmx.de (C.R.); kern@imr.uni-hannover.de (P.K.); kaestner@imr.uni-hannover.de (M.K.); sekretariat@imr.uni-hannover.de (E.R.)

**Keywords:** fringe projection profilometry, harsh conditions, hot measurement object

## Abstract

Optical 3D geometry reconstruction, or more specific, fringe projection profilometry, is a state-of-the-art technique for the measurement of the shape of objects in confined spaces or under rough environmental conditions, e.g., while inspecting a wrought-hot specimen after a forging operation. While the contact-less method enables the measurement of such an object, the results are influenced by the light deflection effect occurring due to the inhomogeneous refractive index field induced by the hot air around the measurand. However, the developed active compensation methods to fight this issue exhibits a major drawback, namely an additional cooling of the object and a subsequent transient illumination component. In this paper, we investigate the cooling and its effect on temporal phase reconstruction algorithms and take a theoretical approach to its compensation. The simulated compensation measures are transferred to a fringe projection profilometry setup and are evaluated using established and newly developed methods. The results show a significant improvement when measuring specimens under a transient illumination and are easily transferable to any kind of multi-frequency phase-shift measurement.

**Dataset License:** CC-BY

## 1. Introduction

The optical 3D geometry measurement [1] of hot a specimen is the content of current research [2] as a subtopic in the context of difficult to measure objects, such as uncooperative surfaces [3]. By this means, different optical geometry reconstruction methods are used. Several authors have used stereo-vision [4,5], light-section [6,7,8,9,10,11,12,13,14,15] or fringe projection profilometry [16] to acquire geometry data for quality control and wear analysis for hot forging processes. In previous papers, we have presented a method to estimate the influence of refractive index gradients on the results of optical 3D geometry measurements of hot specimen by evaluation of the reconstruction quality [17]. Additionally, we have discussed the use of a forced air flow similar to the air knife of [18] to compensate the light deflection effect when measuring the 3D geometry of red-hot specimens [19]. The validity of the compensation was evaluated using the developed reconstruction quality. However, the found improvement of the reconstruction quality metric was considerably smaller than expected based on the results of the background-oriented schlieren (BOS) method and measurements of cold objects. An additional influence was identified as a transient illumination component due to the cooling of the object by the forced air flow. While a change in illumination has been the content of research, their focus laid within the prevention of saturation during image acquisition [20] or in the estimation of a signal-to-noise ratio [21]. Others have investigated the influence of a gamma distortion of the projected intensities [22,23] or an instability of the light source [24] on the quality of a phase-shift measurement.

In this paper, we investigate the transient illumination component occurring during the forced air flow cooling of a hot measurand using pixel intensities and thermal imaging. Its effect on the phase reconstruction using a multi-frequency phase-shift algorithm [25] is analyzed theoretically and in a 1D model setup. The results from both the simulation and the theoretical analysis are discussed and a new image projection sequence is proposed in order to overcome the influence of the forced cooling of the object. Additionally, the concept of an intensity approximation along the acquired projection sequence is shown. In order to compensate the air flow induced intensity transient, the transient is approximated with the help of pixel intensity information in the projection sequence.

The second part of the paper contains an extensive analysis of the newly proposed projection sequence. In order to reveal the positive effect of the proposed projection sequence, a synthetic intensity gradient is incorporated into the projected images. The effect of the new projection sequence as well as of the compensation by approximation is analyzed in this simplified experiment. Additionally, the shown concepts are used for a geometry measurement of a hot object with the added cooling of the forced flow actuator. For the experiment with the hot object under approximately industrial conditions, the new projection sequence is enhanced for improved robustness. This experiment is analyzed w.r.t. the acquired reconstruction quality. The acquired results are discussed and conclusions are drawn.

## 2. Materials and Methods

In this paper, a fringe projection system similar to [26] is used. The applied methods to reconstruct the phase from a sequence of sinusoidal images are discussed in this section. While the origins of the phase reconstruction algorithms lay within the estimation of relative movement from a set of interferometric fringe images [27], this section will only deal with the methods that are used in fringe projection profilometry (FPP). Here, temporal phase unwrapping is common [28,29] while spatial phase calculation methods are used to fill holes in the acquired phasemaps from adjacent phase values. While both methods are interesting and well researched, this paper focuses on temporal phase-shift.

### 2.1. Temporal Phase-Unwrapping

A recent review of temporal phase unwrapping techniques is given by Zou et al. [30]. The authors compared so called multi-wavelength (e.g., [31]), multi-frequency (e.g., [25]) and number-theoretical (e.g., [32]) algorithms. This section focuses on the multi-frequency phase-shift method, since this is used in the setup.

Generally, the intensity $I_{n}\left(u\right)$ of a pixel $u={\left(u,v\right)}^{T}$ projected sinusoidal image *n* can be described [25] as
(1)$$I_{n}\left(u\right)=I_{a}\left(u\right)+I_{p}\left(u\right)I_{r}\sin\left(f\phi +\Delta \varphi_{n}\right),$$
with $I_{a}\left(u\right)$ being the average intensity (or steady component) of the pattern of a pixel u. $I_{p}$ denotes the projected intensity while $I_{r}$ describes the reflectivity. Specific to the projected pattern are its frequency *f* and its phase ϕ, while $\Delta \varphi_{n}$ denotes the phase-shift.

The phase ϕ of a sequence *n* of *N* acquired sine signals can be estimated from three or more measurements of known phase-shifts $\Delta \varphi_{n}$, according to
(2)$$2\pi \;\phi \left(u\right)=\arctan_{2}\frac{\sum_{n=0}^{N-1}I_{n}\left(u\right)\;\sin\left(\Delta \varphi_{n}\right)}{\sum_{n=0}^{N-1}I_{n}\left(u\right)\;\cos\left(\Delta \varphi_{n}\right)}.$$

Using an N=4,n∈{0,1,2,3} pattern sequence with $\Delta \varphi_{n}=0.5\;n\;\pi$ simplifies Equation (Equation 2) to
(3)$$2\pi \;\phi \left(u\right)=\arctan_{2}\left(\frac{I_{0}\left(u\right)-I_{2}\left(u\right)}{I_{1}\left(u\right)-I_{3}\left(u\right)}\right).$$

While these four patterns are sufficiently accurate for an ideal system with an unlimited resolution, its accuracy is limited for a limited resolution, e.g., when using an 8-bit camera sensor. For this case, Peng [25] proposed the use of multiple frequencies *K*, resulting in fringe images with smaller wavelengths. The frequencies $f_{k}$ are calculated according to
(4)$$f_{k}={\left(f_{base}\right)}^{k},$$
with k∈{0,1,...,K−1}. The reconstructed phase $\phi_{k}$ for the images of the higher frequencies k>0 are wrapped, i.e., $-0.5<\phi_{k}<0.5\;\mathrm{for}\;k>0$. Using the integer fringe order m(u), the overall phase ϕ(u) can be determined through
(5)$$f\;\phi \left(u\right)=m\left(u\right)+\phi_{k}\left(u\right).$$

Calculating the fringe order is part of a process called phase unwrapping. It is a hierarchical process, estimating $m_{k+1}\left(u\right)$ through both wrapped phasemaps $\phi_{k}\left(u\right),\phi_{k+1}\left(u\right)$ and the quotient between the frequencies $f_{k+1}/f_{k}$, according to
(6)$$m_{k+1}\left(u\right)=\mathrm{round}\left(\phi_{k}\left(u\right)\;f_{k+1}/f_{k}-\phi_{k+1}\left(u\right)\right).$$

This requires full phase-shift sequences being projected for each frequency $f_{k}$. Using and knowing the average intensity $I_{a}\left(u\right)$ [25] would reduce Equation (Equation 3) to
(7)$$2\pi \;\phi_{k}\left(u\right)=\arctan_{2}\left(\frac{I_{k,0}\left(u\right)-I_{a}\left(u\right)}{I_{k,1}\left(u\right)-I_{a}\left(u\right)}\right).$$

$I_{a}^{\prime }\left(u\right)$ can be estimated from a four-step phase-shift, according to
(8)$$\begin{array}{cc} I_{a}^{\prime }\left(u\right) & =\frac{1}{4}\left[I_{k,0}\left(u\right)+I_{k,1}\left(u\right)+I_{k,2}\left(u\right)+I_{k,3}\left(u\right)\right] \end{array}$$
(9)$$\begin{array}{cc} \; & =\frac{1}{2}\left[I_{k,0}\left(u\right)+I_{k,2}\left(u\right)\right] \end{array}$$
(10)$$\begin{array}{cc} \; & =\frac{1}{2}\left[I_{k,1}\left(u\right)+I_{k,3}\left(u\right)\right]. \end{array}$$

This reduces the required number of images from 4K to 2(K−1)+4, since only one four-step sequence is required to calculate the average intensity $I_{a,k}^{\prime }\left(u\right)$. To get the best sensitivity for the calculation of $I_{a}^{\prime }\left(u\right)$, the highest frequency projection images are used, so $I_{a}^{\prime }\left(u\right)=f\left(I_{k=\max.,n}\left(u\right)\right)$.

### 2.2. Reconstruction Quality Metric

In a previous paper, we have presented a reconstruction quality metric for optical geometry measurements under the influence of a refractive index gradient based on the deviation of corresponding 3D points [17]. For the calculation of the quality metric, a setup with four cameras $n_{c}=4;c\in \left\{0,1,2,3\right\}$ and one projector $n_{p}=1;p=0$ is used. A multi-frequency phase-shift sequence is projected to acquire the projector pixel as a function of the camera pixel, i.e., $u_{p}=f\left(u_{c}\right)$. A virtual raster (cf. [33,34]) is used through the calculation of $u_{c}=f\left(u_{p}\right)$ to find corresponding stereo pairs for all used optical devices [34]. 3D points $x=f(u_{p})$ are triangulated from all stereo pairs *m* and the deviations of corresponding 3D points are used as quality metric to relatively describe the magnitude of the refractive index influence, according to
(11)$$E_{m}\left(u_{p}\right)=\frac{1}{m-1}\sum {\left|x_{i,j}^{\prime }\left(u_{p}\right)-x_{m}^{\prime }\left(u_{p}\right)\right|}_{2}\;\mathrm{for}\;i,j\in m.$$

In the above equation, $x_{m}^{\prime }\left(u_{p}\right)$ denotes the average of all reconstructed 3D points $x_{i,j}^{\prime }\left(u_{p}\right)$ corresponding to that projector pixel $\left(u_{p}\right)$. The validation of the quality metric was shown through the measurement of a discrete refractive index influence in form of a perspex window. The method was used to show the influence of an inhomogeneous refractive index field induced by a hot object on the optical 3D geometry measurement of the object.

### 2.3. Compensation Method for the Light Deflection Effect Occurring in the Optical Geometry Measurements of Hot Objects

In reference to the reconstruction quality metric, we presented a setup for compensating the light deflection effect when measuring the geometry of hot specimen [19]. The setup consists of an additional forced air flow actuator based on nozzles and pressurized air to suppress the inhomogeneous refractive index field developing around a hot object. While the effect was validated qualitatively using the background-oriented schlieren (BOS, cf. [35]) method, the achieved reconstruction quality for a measurement without time delay was estimated to be worse than the original quality when measuring a cold object. A temporal intensity transient was observed during the acquisition of the projection sequence. This transient intensity was considered to be an additional influence on the reconstruction quality.

### 2.4. Simulation Model

A simplified model is used to estimate the influence of a transient illumination on the phase reconstruction using a multi-frequency phase-shift algorithm. Here, the phase 0≤ϕ≤1 of a multi-frequency phase-shift sequence is calculated through Equations (Equation 2) and (Equation 7) from nominal values of 1D sinusoidal function. This simulation uses unitless expressions comparable to the representation of an image as matrix of 8-bit integers. To introduce a transient illumination, a gradient value is added to the single sine values. This gradient value is modeled as a function of the images index, which is corresponding to a temporal background illumination gradient in the fringe projection measurements. The results are compared to a nominal phase value by an uninfluenced reconstruction. The sine functions have a nominal amplitude of $I_{p}\;I_{r}=100$ with a nominal offset of 100, i.e., lie between 0 and 200, and the intensity values are discretized to integer numbers.

### 2.5. 3D Measurement Setup

The full 3D measurement setup comprises four cameras (AV Prosilica GT), one projector (WinTech PRO4500) and one thermographic camera (InfraTec Vario-CAM HD—infrared (IR) wavelength range (7.5–14) μm—not shown) and is depicted in Figure 1. The distance from the optical center of the projector to the measurement object is approx. 400 mm. The forced air flow is created by an actuator consisting of a pressure regulator, a magnetic valve and a set of nozzles in an aluminum block. In the current setup, an 8-image and a 12-image unwrapping algorithm are used with three frequencies $f=6^{k}$ each. An extensive description of the physical setup is beyond the scope of the paper. The interested reader is referenced to [17,19].

## 3. Influence of the Forced Air Flow on Phase-Shift Measurements

To analyze the effect of the reduction of radiation from self-emission during cooling, the acquired images during a full-field illumination of the specimen are examined. The exposure time is chosen to be 80 ms, while the projection frequency is set to 10 Hz. The forced flow actuator is activated for 5 s at $t_{c}\approx 1.3\;s$. The mean intensity development of a rectangular part of the image is shown for one of the cameras in Figure 2b. Additionally, the radiation from self-emission of the specimen is shown as measured by the thermal imaging camera (Figure 2b dashed black lines). The development of the intensity difference is following the radiation from self-emission closely. The development of both the IR data and the camera intensity data is considered a quadratic form for $1.3\;s<t_{c}<3.0\;s$ and a linear form for $3.0\;s<t_{c}<6.0\;s$. After the air flow is deactivated at $t_{c}\approx 6.3\;s$, both the IR intensity and the camera intensity increase.

### 3.1. Theoretical Considerations

Assuming a linear intensity gradient $\Delta {\dot{I}}_{t}\left(u\right)=\Delta I_{t}\left(u\right)\;/\;\Delta t=\mathrm{const}.$ and a time period between the image acquisition of $I_{0,k}$ and $I_{2,k}$ of $\Delta t_{0,2}$ and between $I_{1,k}$ and $I_{3,k}$ of $\Delta t_{1,3}$, then the phase calculation (cf. Equation (Equation 3)) is changed to
(12)$$2\pi \;\phi_{k}\left(u\right)=\arctan_{2}\left(\frac{I_{0,k}^{*}\left(u\right)-I_{2,k}^{*}\left(u\right)-\Delta {\dot{I}}_{t}\left(u\right)\;\Delta t_{0,2}}{I_{2,k}^{*}\left(u\right)-I_{3,k}^{*}\left(u\right)-\Delta {\dot{I}}_{t}\left(u\right)\;\Delta t_{1,3}}\right).$$

In this equation $I_{n,k}^{*}\left(u\right)$ denotes the uninfluenced intensity without the gradient at the image with index *n*. Assuming a sequence with N=4 and $\Delta \varphi_{n}=0.5\;n\pi$ and using a projector with an equidistant projection interval, the intensity gradients $\Delta {\dot{I}}_{t}\left(u\right)\;\Delta t_{0,2};\Delta {\dot{I}}_{t}\left(u\right)\;\Delta t_{1,3}$ can be simplified to
(13)$$\Delta {\dot{I}}_{t}\left(u\right)\;\Delta t_{0,2}=\Delta {\dot{I}}_{t}\left(u\right)\;\Delta t_{1,3}=\Delta {\dot{I}}_{i}\left(u\right)\;\Delta t_{i}.$$

In this expression, $\Delta t_{i}$ denotes the integer time interval between two projection as number of images while $\Delta {\dot{I}}_{i}\left(u\right)$ is the change of intensity normalized to one interval period. The phase calculation the then simplified to
(14)$$2\pi \;\phi_{k}\left(u\right)=\arctan_{2}\left(\frac{I_{0,k}^{*}\left(u\right)-I_{2,k}^{*}\left(u\right)-\Delta {\dot{I}}_{i}\left(u\right)\;\Delta t_{i}}{I_{1,k}^{*}\left(u\right)-I_{3,k}^{*}\left(u\right)-\Delta {\dot{I}}_{i}\left(u\right)\;\Delta t_{i}}\right).$$

For the regular 12-image and high-frequency 8-image sequence (cf. Equation (Equation 3)), both with constant phase-shifts of $\Delta \varphi_{n}=0.5\;n\pi$, this interval is constant $\Delta t_{i}=2$.

For the low-frequency 8-image sequence, Equation (Equation 7) under the influence of an intensity gradient $\Delta {\dot{I}}_{i}\left(u\right)$ is changed to
(15)$$2\pi \;\phi_{k}\left(u\right)=\arctan_{2}\left(\frac{I_{0,k}^{*}\left(u\right)-I_{a}^{*}\left(u\right)-\Delta {\dot{I}}_{i}\left(u\right)\;\Delta t_{t,a,0}}{I_{1,k}^{*}\left(u\right)-I_{a}^{*}\left(u\right)-\Delta {\dot{I}}_{i}\left(u\right)\;\Delta t_{i,a,1}}\right).$$

In this case, the index differences are not equal $\Delta t_{t,a,0}\neq \Delta t_{t,a,1}$, since these index differences are corresponding to the difference between the virtually fixed position of $I_{a}$ and the changing index of $I_{0,k}^{*}$ and $I_{1,k}^{*}$, respectively

### 3.2. Discussion and Conclusions from the Observations

The observations from the use of the forced air flow actuator show a darkening of the captured images over time, which is similar to the reduction of radiation from self-emission caused by the cooling of the object through the air flow (cf. Figure 2). Due to this similarity, it is assumed that the reduction of radiation is causing the intensity reduction in the camera images. For this consideration, the source of the radiation reduction is irrelevant, i.e., there is no difference between the effect of a changing emission coefficient and a thermal cooling. Due to the influence of a background illumination gradient on the phase reconstruction of a fringe projection system, it is necessary to reduce it. There are two possible solutions to this challenge:reduce the influence of $\Delta {\dot{I}}_{i}\left(u\right)\Delta t_{i}$ on the phase reconstruction, orcompensate for the intensity gradient through an estimation.

To reduce the absolute intensity difference between two subsequent images (1.), it is proposed to change the order in which the patterns are projected. Projecting the images that exhibit a phase-shift of Δφ=π in direct succession is assumed to reduce $\Delta {\dot{I}}_{i}\left(u\right)\Delta t_{i}$ by dividing $\Delta t_{i}$ by half. The regular and proposed adapted image sequences are displayed in Table 1.

To compensate for the intensity transient, it is necessary to estimate its magnitude in a fringe projection sequence. By this means, Equations (Equation 8)–(Equation 10) can be used, taking advantage of the π phase-shift images. This results in three estimates per full phase-shift sequence at different points in time $n_{m}$. A schematic depicting the usage of different images to estimate $I_{a}^{\prime }\left(u\right)$ is shown in Figure 3. The acquired pixel-wise steady intensities $I_{a}^{\prime }\left(u\right)$ and corresponding images indices $n_{m}$ are used to calculate functional parameters p estimate the intensity as $I_{est}\left(u\right)=f\left(p,n,u\right)$.

### 3.3. Results from the Simulation Model and Discussion

The influence of a temporal intensity gradient in a sequence of multi-frequency phase-shift images is examined using the simulation model. By this mean, different gradient functions are modeled (see Figure 4a). For the linear gradient, the average intensity $I_{a,\mathrm{linear}}^{*}$ is modeled as
(16)$$I_{a,\mathrm{linear}}^{*}\left(i\right)=100-2i,$$
while the quadratic influence is approximated by
(17)$$I_{a,\mathrm{quadratic}}^{*}\left(i\right)=0.5i^{2}-2i-40.$$

The projection magnitude is kept constant at $I_{p}=100$. These values for the influence are chosen arbitrarily. The results for the linear influence are shown in Figure 4b,c while the results for the quadratic influence are shown in Figure 4d. Figure 4b shows a periodic difference of the influenced phase compared to the nominal phase. Its frequency is 36, which equals the highest frequency of the projected patterns $f_{k}=36$. This is supported by the depiction of one and a half full oscillations for 1/36≤ϕ≤2.5/36 (see both Figure 4c,d). The magnitude of the difference to the reference phase is in the range of $10^{-4}$ for an influence of $\Delta {\dot{I}}_{i}/I_{p}=0.02$. The adapted sequence is reducing the difference to the reference phase by approximately half for both the 12-image sequence and the 8-image sequence. There is no influence of the images with k∈{0,1} observable in the simulation model due to the rounding operation in Equation (Equation 6). The phase information itself is only stored in the k=2, i.e., $f_{k}=36$, images while the lower frequency images are used for the unwrapping operation. Therefore, there is no additional benefit of using the 12-image sequence compared to the 8-image sequence, since the greater magnitude of $\left(\Delta \dot{I}\;\Delta t_{a,0}\right)$ and $\left(\Delta \dot{I}\;\Delta t_{a,1}\right)$ compared to $\left(\Delta \dot{I}\;\Delta t_{i}\right)$ has no influence on the phase calculation itself.

For the stated reasons, the difference to the nominal phase is only a function of the intensity transient in the highest projected frequency. This is observable through the comparison of the differences in Figure 4 in context with the modeled influences: the difference magnitudes to the nominal phase values are smaller for the quadratic influence compared to the linear influence even though the overall intensity gradient for the quadratic influence is greater compared to the linear influence (see Figure 4a). This is explained through the lower intensity gradient of the quadratic influence during the highest frequency projection (k=2 and image index 0≤n≤3).

## 4. Experiments

### 4.1. Experimental Setup

Due to the shown significance of the intensity gradient on the phase calculation, the proposed solutions need to be examined. Both are separately analyzed using a synthetic intensity gradient induced into the projection sequence in the hardware setup. Using 8-bit projection images, the amplitude is set to $I_{p,\mathrm{grad}}=100$ and the intensity gradient is set linearly to $\Delta {\dot{I}}_{i}=2$ per image, i.e., the steady component for image n=0 is at $I_{a,n=0}=100$, for image n=1 at $I_{a,n=1}=102$, etc. The values ($I_{p};\Delta \dot{I}$) for this synthetic intensity gradient are chosen corresponding to the gradient in the simulation (cf. Equation (Equation 16)). The sequence is projected onto a white piece of paper and the calculated phase is compared to a reference sequence using the 8-image unwrapping and the maximum possible amplitude of $2I_{p,\mathrm{ref}}=255$. The phase evaluation is shown as a 1D section from the green line from Figure 5a to enable a comparison in the same dimension as the simulation results (cf. Figure 4). To estimate the background phase noise, there is also a projection sequence with an amplitude of $I_{p,\mathrm{noise}}=100$ and no added intensity gradient $\Delta {\dot{I}}_{i,\mathrm{noise}}=0$.

To analyze the effect of an intensity-gradient compensation based on an intensity estimate (cf. (2.) from Section 3.2), the same images acquired for the above analysis are used. The intensity-gradient compensation is based on a linear approximation using Equations (Equation 8)–(Equation 10) to calculate p(u) and therefore $I_{n,\mathrm{est}}\left(u\right)$ and its temporal component ${\dot{I}}_{n,\mathrm{est}}\left(u\right)$. The acquired parameters p(u) are then used to calculate the estimated intensities without gradient $I_{n,\mathrm{comp}}^{*}\left(u\right)$, according to
(18)$$I_{n,\mathrm{comp}}^{*}\left(u\right)=I_{n}\left(u\right)-{\dot{I}}_{n,\mathrm{est}}.$$

$I_{n,\mathrm{comp}}^{*}\left(u\right)$ is then used to calculate the estimated uninfluenced phase value $\varphi_{n,\mathrm{comp}}^{*}$ using the presented regular algorithms.

The presented methods are also evaluated in the design application, i.e., when measuring hot components. By this means, the compensation method for the light deflection effect [19] is used and the presented methods are applied in order to shorten the gap w.r.t. the reconstruction quality metric between a cold reference measurement and a hot measurement under the influence of a forced air flow. To investigate the change in the reconstruction quality metric, the 8-image algorithm is used with and without adapted sequences. In addition, the sequences are used to estimate the intensity gradient as a linear function as well as a quadratic function.

### 4.2. Sequence Analysis and New Adapted Projection Sequence

While the intensity estimation from the used regular sequences has not yet shown to be difficult, the necessary extrapolation of the intensity to compensate the gradient in the outer parts of the sequences, i.e., image indices n∈{0,3}, might be an issue. As shown in Figure 3, the intensity for the images with $f_{k}=\max.$ and n∈{0,3} are extrapolated from $n_{m}\in \left\{1,\;1.5,\;2\right\}$ and $n_{m}\in \left\{0.5,\;1.5,\;2.5\right\}$, respectively. To eliminate this extrapolation as an error source, it is proposed to use the enhanced adapted sequence from Table 2.

In our opinion, the enhanced adapted sequence offers a solution to the trade-off between the added measurement time through the addition of new projection images and the expected quality of the phase measurement. It contains the same number of images as a regular 12-image sequence, but offers a high number of support points for the intensity estimation, as the average intensity can be calculated from a rolling average through the images i∈{2,3,...,9}. Compared to the 8-image algorithm and the 12-image algorithm which both feature three high-sensitivity supporting points, the intensity estimation (cf. Equations (Equation 8)–(Equation 10)) for the enhanced adapted sequence can be based on nine supporting points (i∈{{2,3},{4,5},{6,7},{8,9},{2,3,4,5},…,{6,7,8,9}}). Additionally, the phase calculation can be based on five separate four-image sequences from i∈{2,3,…,9}. It is expected to acquire more robust phase measurements from the rolling calculation through the availability of five separate phase reconstruction results. These single results can be evaluated statistically, e.g., by calculating the average phase value $\phi_{m}$ for each pixel.

### 4.3. Experimental Results

The reduction of the influence of the intensity transient through the use of the adapted sequence as well as the compensation of such a transient through intensity approximation are evaluated using different methods. The adapted sequence is evaluated under the influence of a deterministic gradient in an experiment with a synthetic intensity transient while the enhanced adapted sequence is evaluated under the influence of a hot measurand in a workshop experiment. The workshop experiments are conducted w.r.t. the delay time between activation of the flow actuator and the start of the measurement. This delay time $\Delta t_{d}$ was previously shown to be an influence on the established reconstruction quality metric [19]. In the present paper, the following delay times are investigated $\Delta t_{d}\in \left\{0.3\;s,1.3\;s,2.3\;s,3.3\;s\right\}$ of which 0.3
s are assumed to be the time between the activation of the flow actuator and the full development of the flow field. Other parameters of the measurements include the geometrical properties of the used cylinder ($d_{c}\approx 50\;mm;l_{c}\approx 250\;mm$) as well as its temperature $\vartheta_{c}\approx 1000\;{}^{\circ }C$. The experiments of the measurement series consist of a measurement of the hot component with a deactivated air flow actuator, subsequently followed by a measurement in which the air flow actuator is activated. The measurement results are structured corresponding to the delay time $\Delta t_{d}$. The measurand is not moved in between the subsequent single measurements to reduce the effect of systematic influences related to the 3D pose and the surface properties of the object on the reconstruction quality. The identical procedure was used in [19] and was discussed in detail in [17].

#### 4.3.1. Synthetic Intensity Gradient

The results for the experiment with the synthetic intensity gradient are shown in Figure 5b for the evaluation of the adapted projection sequence and in Figure 5c for the compensation of the intensity gradient through estimation. In both cases, the results from the 8-image reconstruction algorithm are depicted, while the results for the 12-image algorithm are not shown, since the results for both algorithms are identical in magnitude and form.

For the evaluation of the adapted sequence (cf. Table 1), the difference between the acquired phase values under the influence of the synthetic intensity gradient to the reference phase is uniformly modulated (cf. Figure 5b). In correspondence to Figure 4 and Figure 5, the frequency of the modulation is estimated to be approximately $f_{k=2}=36$. Similar to the results from the simulation model, the use of the adapted sequence reduces the phase error by approximately a factor of 0.5 in the experiment.

Examining the results of the intensity-gradient compensation by estimation (see Figure 5c), both used sequences show an error in magnitude identical to the magnitude of the background phase noise. Therefore, the phase error of the intensity-gradient compensation is assumed to be mainly due to the phase reconstruction noise.

#### 4.3.2. Intensity Estimation Evaluation

The following results were acquired in the experiment with the hot cylinder in the workshop. For these experiments, the quality of the intensity estimation is evaluated on a per-pixel basis using the variance of the average intensity $I_{a,\mathrm{comp},n_{m}}^{*}\left(u\right)$ after calculating $I_{n,\mathrm{comp}}^{*}\left(u\right)$ for the chosen adaptation method, i.e.,
(19)$$q_{i}\left(u\right)=\sqrt{\frac{1}{N_{m}}\sum_{N_{m}}{\left(I_{a,\mathrm{comp},n_{m}}^{*}\left(u\right)-I_{a,\mathrm{comp},m}^{*}\left(u\right)\right)}^{2}},$$
with the number of intensity estimates $N_{m}$. For the presented case, $I_{a,\mathrm{comp},m}^{*}\left(u\right)$ is calculated as $I_{a,\mathrm{comp},n_{m}}^{*}\left(u\right)$, according to
(20)$$I_{a,\mathrm{comp},m}^{*}\left(u\right)=\frac{1}{N_{m}}\sum_{N_{m}}I_{a,\mathrm{comp},n_{m}}^{*}\left(u\right).$$

This approximation quality $q_{i}\left(u\right)$ is depicted in Figure 6 for different kinds of approximation functions and delay times $\Delta t_{d}\in \left\{0.3\;s,1.3\;s\right\}$. The initial magnitude without compensation of the intensity variance $q_{i}$ appears to be larger for $\Delta t_{d}=0.3\;s$ compared to $\Delta t_{d}=1.3\;s$. The approximation quality seems to be insensitive to a change of the delay time, as the difference in $q_{i}\left(u\right)$ between $\Delta t_{d}=0.3\;s$ and $\Delta t_{d}=1.3\;s$ is insignificant when using the same type of approximation. Comparing the approximation quality $q_{i}\left(u\right)$ within one delay time shows slightly lower variance values for the linear approximation compared to the quadratic approximation.

#### 4.3.3. Phase Variance Evaluation

For the proposed adapted sequence from Table 2, the quality of the phase calculation can be estimated using the variance of the rolling phase reconstruction $q_{\phi }$, using the increased number $N_{\phi }$ of phase calculations $\varphi_{n_{\phi }}$ and their indices $n_{\phi }\in \left\{0,1,...,N_{\phi }-1\right\}$. This phase reconstruction quality $q_{\phi }$ is assumed to be proportional to the quality of the intensity gradient compensation, since a change in average illumination is causing a gradually changing the phase calculation. This will result in a higher phase variance compared to the phase measurement under constant ambient illumination. The phase variance qφ(u) is depicted in Figure 7 and is calculated according to
(21)$$q_{\phi }\left(u\right)=\frac{1}{N_{\phi }}\sum_{n_{\phi }}{\left(\phi_{n_{\phi }}\left(u\right)-\phi_{m}\left(u\right)\right)}^{2},$$
using the average phase value $\phi_{m}$. The calculated phase reconstruction quality $q_{\phi }$ appears to be similar for both investigated delay times $\Delta t_{d}\in \left\{0.3\;s,1.3\;s\right\}$ and is significantly decreased by the use of any of the proposed approximation methods. There seems to be no significant difference in between the results for both approximation methods.

#### 4.3.4. Reconstruction Quality Estimation

The direct influence of the intensity gradient compensation on the reconstruction of 3D geometry can be analyzed using the previously established reconstruction quality estimation based on the variance of corresponding 3D points using multiple cameras [17]. The reconstruction quality is calculated on a per-pixel basis and is summarized using the median of all valid measured points for an adequate and easy comparison in between different adaption methods and different delay times. This is also an adaptation due to the challenges which appeared during the experiments for the earlier papers [17,19], in which the color and surface of the measurement object have an effect on the per-pixel for the established reconstruction quality estimation. The results are depicted as bar diagrams in Figure 8.

The acquired results show similar median deviations for all four measurements (Figure 8, light colors, sparsely hatched area) of the hot object ($\vartheta_{c}\approx 1000\;{}^{\circ }C$) without the forced air flow and without the compensation of the intensity transient through approximation. Similar to prior experiments [19], the reconstruction quality is estimated at lower levels (higher deviations) when activating the forced flow actuator for $\Delta t_{d}<3.3\;s$.

## 5. Discussion

In this section, the acquired results of the experiments are discussed. This section is separated into subsections corresponding to the subsection from Section 4 for clarity reasons.

### 5.1. On the Results of the Synthetic Gradient Experiment

The results from the synthetic intensity gradient experiment (see Figure 5) show a similar effect of the intensity gradient on the phase reconstruction as the previously discussed simulation model (see Figure 4). The remaining phase error in the experiment was shown to be induced by the background phase noise. Therefore, the simplification of the phase calculation in the simulation to a 1D calculation can be considered valid. The use of the adapted sequence without intensity approximation comes at no additional costs of computation or acquisition time.

The acquired results for the compensation by intensity approximation show a significant reduction of the systematic phase error down to the magnitude of the phase noise (see Figure 5c) for a linear illumination transient. Therefore, it can be concluded that the estimation of the temporal intensity transient compensates for its effect for a synthetic linear gradient. From this experiment, no conclusions can be drawn regarding the non-linear transients occurring during the cooling of a hot object (cf. Figure 2). The costs of the approximation are related to the additional computation time for the intensity approximation.

### 5.2. On the Results of the Approximation Quality Evaluation

Surprisingly, the linear approximation results in a lower intensity variance value $q_{i}\left(u\right)$ compared to the quadratic approximation while having a lower number of variables to optimize. Both methods exhibit neglectable variances compared to the original images. However, a conclusion on the improvement of the quality of the geometry reconstruction based on the improvement of the intensity approximation quality is invalid. The validity of the intensity variance as a quality metric for the intensity approximation is evaluated by comparing Figure 2 and Figure 6: The intensity gradient during the air flow induced cooling is steepest in the first seconds after the activation of the air flow. Therefore, the intensity variance without intensity estimation is expected to be larger for $\Delta t_{d}=0.3\;s$ compared to $\Delta t_{d}=1.3\;s$. This expectation is supported by the data in Figure 6.

### 5.3. On the Results of the Phase Reconstruction Quality

The quality of the intensity estimation can also be assessed using the phase variance of the enhanced adapted sequence via the rolling phase calculation (see Figure 7). This assessment also shows a reduction of the variance using the intensity approximation. A difference between the linear and the quadratic estimation cannot be concluded from the shown phase variance, neither can a difference between $\Delta t_{d}=0.3\;s$ and $\Delta t_{d}=1.3\;s$. Therefore, the intensity variance $q_{i}$ and the phase variance $q_{\phi }$ are suitable as an auxiliary quality estimation method to describe the influence of an intensity approximation on the 3D phase-shift measurement under changing illumination conditions. However, neither the approximation quality nor the phase reconstruction quality can be used as singular indicators for an improvement of 3D geometry reconstruction quality by using the proposed method of intensity approximation.

### 5.4. On the Results of the Quality Estimation of the Geometry Reconstruction

The direct estimation of the reconstruction quality using the previously established metric [17] shows a general improvement of the reconstruction quality $E_{m}$ when using the enhanced adapted projection sequence in combination with an intensity adaptation compared to the regular sequence without forced air flow. The largest improvement from the linear approximation to the quadratic approximation is observed in the measurement with $\Delta t_{d}=0.3\;s$ (red). The reason for this appears to be the non-linearity of the intensity gradient in the allocated time slot. A similar effect with a lower magnitude is observed for the measurement with $\Delta t_{d}=1.3\;s$ (green). This improvement of the geometry reconstruction quality was not expected from the results in Figure 6 and Figure 7, which show insignificant differences between $\Delta t_{d}=0.3\;s$ and $\Delta t_{d}=1.3\;s$ and between the linear approximation and the quadratic approximation.

There is no further improvement of the geometry reconstruction quality $\mathrm{median}(E_{m})\approx 0.15\;mm$ observable in the measurements for $\Delta t_{d}\geq 2.3\;s$, indicating a mostly linear intensity gradient from this delay time on. The minimal deviation level for the measurement of the hot objects ($\vartheta_{c}\approx 1000\;{}^{\circ }C$) is reached using the quadratic approximation with $\Delta t_{d}=1.3\;s$. Larger delay times do not result in smaller reconstruction deviations $E_{m}$ for the hot measurements while not quite reaching the level of the cold reference measurements (blue).

While the majority of the results follows the systematics described above, some are inconclusive. The median deviation for the intensity compensation through linear approximation for a deactivated flow actuator and $\Delta t_{d}=0.3\;s$ is smaller compared to any other except the cold reference measurement. Additionally, the median deviation for the deactivated flow actuator without intensity compensation for $\Delta t_{d}=2.3\;s$ is significantly smaller than the deviation for the other delay times with otherwise identical parameters. The validity of these values is in doubt, since the earlier paper [19] showed a significant light-deflecting refractive index field around the measurement object for cases using a deactivated flow actuator. However, since the reconstruction deviation $E_{m}$ can only be interpreted statistically, there are several effects leading to lower values of $E_{m}$ (cf. [17]), e.g., angle between light ray and boundary of the refractive index field as well as improving visibility of fringes on the lighter colored surface of the wrought-hot object.

The limits of the intensity transient compensation through approximation appear to be mainly due to the used approximation function. The linear approximation fails to produce high quality geometry reconstruction results for $\Delta t_{d}\leq 1.3\;s$ and especially for $\Delta t_{d}=0.3\;s$. In contrast, the compensation by quadratic approximation shows median deviation values of $\mathrm{median}(E_{m})\approx 0.15\;mm$ for any of the examined delay times $\Delta t_{d}$. For a closer examination, finer increments of $\Delta t_{d}$ are needed. Additionally, the use of incremental spline interpolation instead of functional approximation might solve high-degree intensity development.

## 6. Summary

In this paper, we have investigated the increased reconstruction deviations when trying to compensate for the light deflection effect caused by a hot measurement object using a forced flow actuator. The intensity gradient caused by the cooling of the object through the air flow was postulated to be the reason for this increase. A synchronized acquisition of grayscale images and thermographic images confirmed this effect. Its influence on the multi-frequency phase-shift measurement was investigated on a theoretical level, in a 1D model simulation as well as in a 2D experiment using a synthetic intensity gradient. From the conclusions of these experiments, a new adapted projection sequence was proposed and implemented. A method for the approximation of the intensity gradient was presented and evaluated. The evaluation was conducted using different methods based on the novel projection sequence and the previous established reconstruction quality metric. The new method did improve the reconstruction quality level of the measurements of the hot object significantly compared to not using a flow actuator and compared to using a flow actuator with a regular projection sequence, but could not reach the level of the cold reference measurement.

## 7. Conclusions

The enhanced adapted projection sequence in combination with the established forced flow actuator and the intensity transient compensation is significantly improving the reconstruction quality when measuring the 3D geometry of hot objects. The main disadvantage of the adapted enhanced sequence is the extension of the necessary projection time $t_{p}$ by 50% and the increase of computation time. While the increase in measurement time is large on a relative scale, the absolute time is increasing from $t_{p}\approx$
1s to $t_{p}\approx$
1.5 s. This projection time can also be decreased using an improved measurement setup with cameras and projectors with higher frame rates. Overall, the benefits from the proposed projection sequence outweigh the drawbacks.

Generally, the adapted sequence with swapped projection image indices (cf. Table 1) can be applied in any usage scenario for a fringe projection profilometry using a multi-frequency phase-shift phase reconstruction. The change order of the projection sequence causes no additional costs and reduces the influence on any occurring intensity transients, whether known or not.

## 8. Future Work

To generalize the results beyond a singular measurement object, a semi-automatic selection of the delay time is needed with the use of a single-camera single-projector FPS system. By this means, the projection of a rolling high-frequency phase-shift sequence can be used in combination with Equation (Equation 19). In the future, a holistic analysis of the optical 3D geometry measurements considering calibration error and expected 3D point noise should be conducted, since both appear to be significant influences on the reconstruction deviation $E_{m}$. A possible road map is to combine a phase noise calibration [36] with a phase noise to 3D reconstruction noise calculation [37].

## Figures and Tables

**Figure 1 sensors-21-01599-f001:**
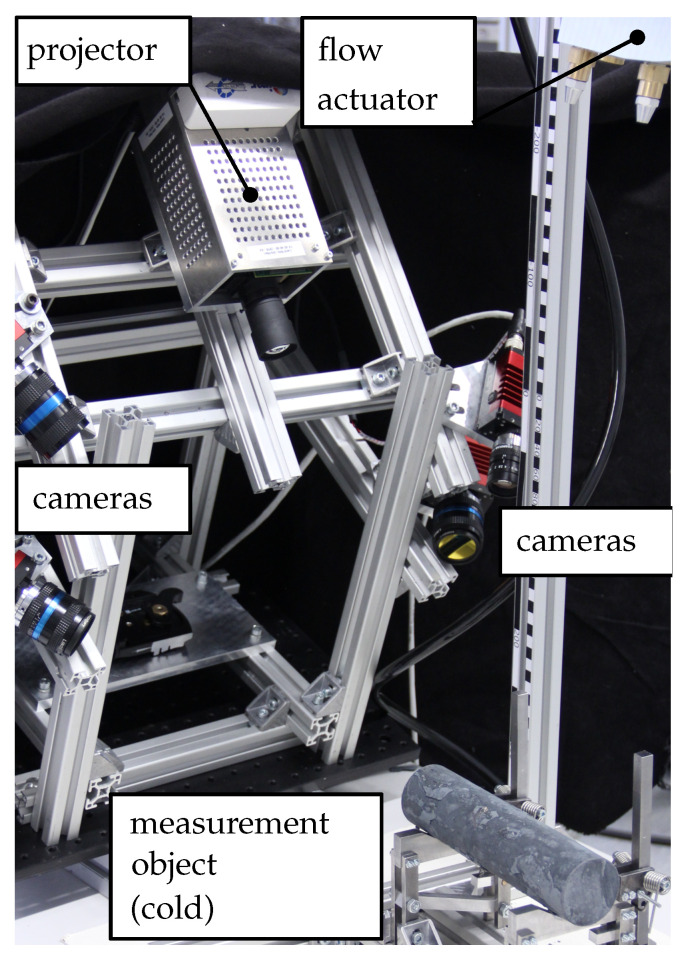
Image of the used 3D measurement setup.

**Figure 2 sensors-21-01599-f002:**
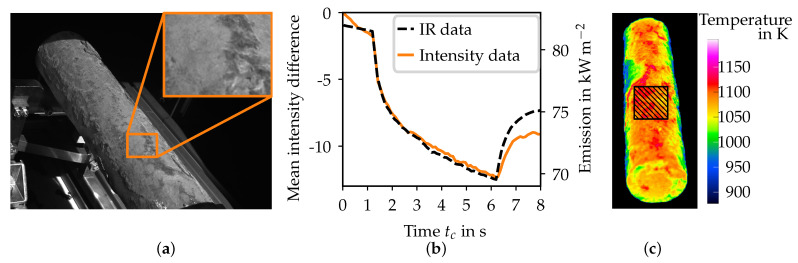
Development of the mean relative pixel intensities and the infrared emission intensity during the activation of the forced flow actuator. (**a**) Exemplary camera image of the used rectangle area on the specimen; (**b**) comparison between the mean acquired intensity from (**a**), corresponding to the left axis and the mean emission intensity from (**c**), corresponding to the right axis; (**c**) calculated temperature distribution assuming an emission coefficient of one from the used thermographic image. The rectangle, where the mean IR data are taken from, is also depicted.

**Figure 3 sensors-21-01599-f003:**
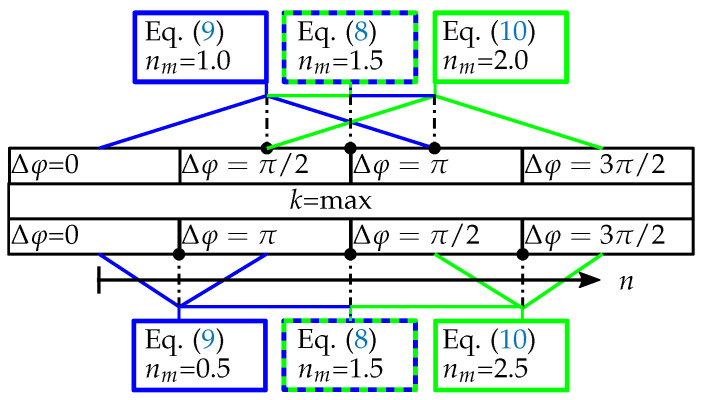
Used images for the estimation of the background intensity $I_{a}^{\prime }\left(u\right)$ for both the regular image sequence (top) and the adapted sequence (bottom). The shown sequence is an extract from the whole sequence focusing on the highest frequency projections k=2.

**Figure 4 sensors-21-01599-f004:**
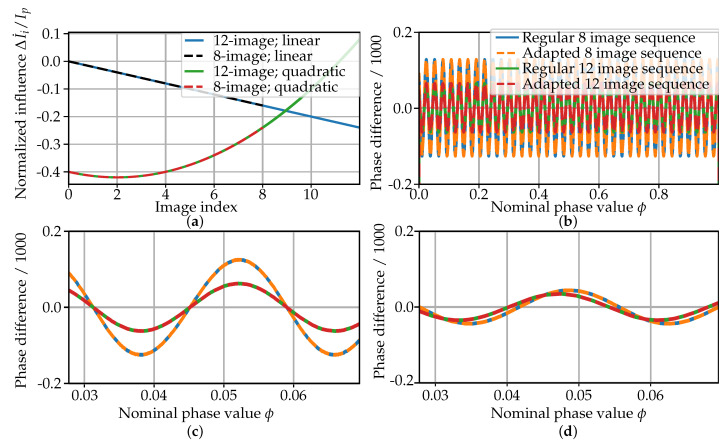
Results of the 1D model experiment. (**a**) Used model influences as a normalized value over image index; (**b**) result for a linear influence over the full range of nominal phase values; (**c**) zoom in on results with the linear influence from 1/36≤ϕ≤2.5/36; (**d**) zoom in on results with the quadratic influence from 1/36≤ϕ≤2.5/36.

**Figure 5 sensors-21-01599-f005:**
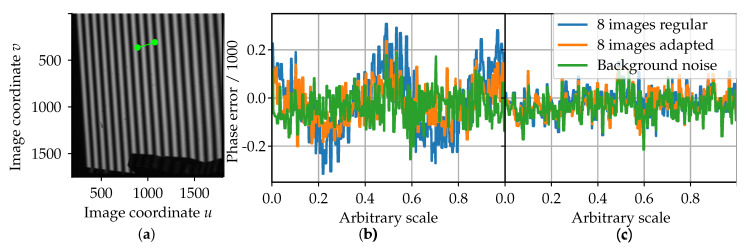
Evaluation of the difference between a measured phase with a sequence influenced by a synthetic intensity gradient and a reference phase. (**a**) Excerpt from a raw camera image of a projection with $f_{k}=36$ and Δφ=0; (**b**) comparison of the phase error for the regular sequence and the adapted sequence; (**c**) phase error for a linear intensity estimation.

**Figure 6 sensors-21-01599-f006:**
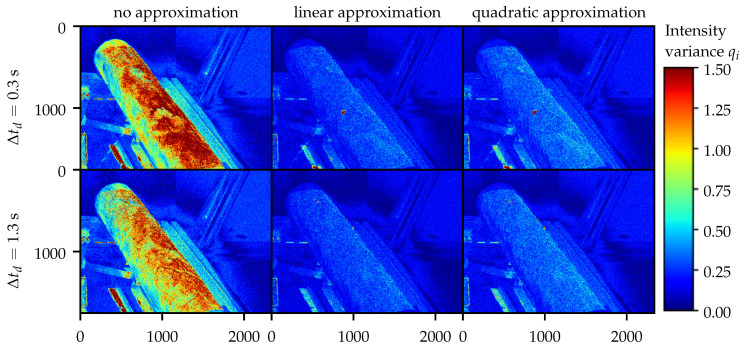
Intensity variance $q_{i}\left(u\right)$ for different delay times $\Delta t_{d}\in \left\{0.3\;s,1.3\;s\right\}$ and different intensity approximation methods. The variance is based on the background intensity calculated from the given support points shown in Equation (Equation 19).

**Figure 7 sensors-21-01599-f007:**
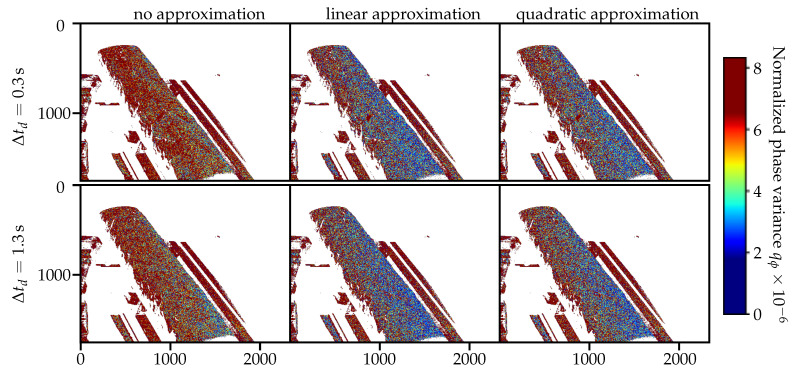
Per-pixel phase variance $q_{\phi }\left(u\right)$ for the rolling phase calculation for different delay times $\Delta t_{d}\in \left\{0.3\;s,1.3\;s\right\}$ and different intensity estimation methods.

**Figure 8 sensors-21-01599-f008:**
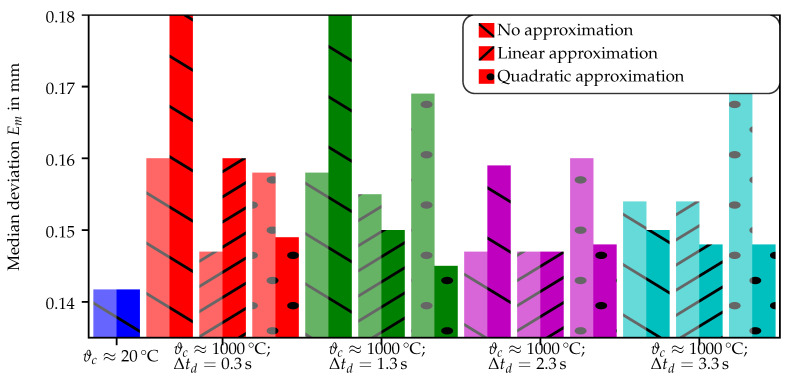
Summary of the acquired reconstruction quality values as the median of the valid pixels. Lighter colors mark the reconstruction value for the measurement without the activation of the flow actuator using a regular 8-image sequence. The saturated bars mark the measurement using the adapted sequence (see Table 2) and the given intensity estimation method.

**Table 1 sensors-21-01599-t001:** Assignment of frequency and phase shifts of the 8-image multi-frequency phase-shift sequence. The adapted sequence is marked with a $\Delta \varphi^{*}$.

Image Index *i*	0	1	2	3	4	5	6	7
frequency exponent *k*	0	0	1	1	2	2	2	2
phase shift Δφ	0	$\frac{\pi }{2}$	0	$\frac{\pi }{2}$	0	$\frac{\pi }{2}$	π	$\frac{3\pi }{2}$
phase shift $\Delta \varphi^{*}$	0	$\frac{\pi }{2}$	0	$\frac{\pi }{2}$	0	π	$\frac{\pi }{2}$	$\frac{3\pi }{2}$

**Table 2 sensors-21-01599-t002:** Proposed enhanced adapted projection sequence for a robust estimation of the intensity gradient.

	Image Index *i*	0	1	2	3	4	5	6	7	8	9	10	11
8-image	frequency exponent *k*	0	0	2	2	2	2	2	2	2	2	1	1
	phase shift Δφ	0	π/2	0	π	π/2	3π/2	0	π	π/2	3π/2	0	π/2

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
