# Peer review of "Adapted Fringe Projection Sequences for Changing Illumination Conditions on the Example of Measuring a Wrought-Hot Object Influenced by Forced Cooling"

_sensors, 2021, doi:10.3390/s21051599_

Round 1

Reviewer 1 Report

This paper investigates the correction of reconstruction deviations induced by the variance of illumination in constructing geometry of a hot object using the fringe projection combined with a forced flow actuator. The swapped sequence and intensity gradient compensation are used to reduce the influence of airflow cooling. The topic falls within the scope of the Sensors. Several points, however, need to be clarified and addressed before publication.

  1. Fig. 2(b) shows the decrease of pixel intensity and the infrared emission intensity during cooling period. Line 136-138 indicates that the variation is in a quadratic form for 1.3s~3s and a linear form for 3s~6s. However, the infrared emission intensity is influenced by the emissive coefficient of the object surface and the type of infrared cameras. What’s the range of wavelength of the infrared camera?

If different materials and surface emission coefficients are used for the inspection, this curve may be varied.

  1. What’s the distance from the 3d measurement setup to the detected object? Does the change of distance influence the illumination and light deflection?

  1. Many symbols are undefined, e.g., line 154, Δia,nom and Δia,den; line 176, Iest(u); in Fig. 3, nm; in Fig. 4(a), Ip; in Eq. 17, Nm.

  1. Be careful about writing, e.g. line 170, “is is”; line 307, “TThe”.

  1. The linear and quadratic compensation is given in Fig. 4(a). However, the mathematical expression of these two functions should be presented in the paper. In line 211, why is the intensity gradient set to 2 per image in the experiment? Discussion is needed since this is important for the precision 3d measurement.

  1. In Fig. 8, please explain why the measurement without the flow actuator using a regular 8-image sequence achieves the smallest median deviation when Δtd=0s? Does it show that the proposed method is not applicable when Δtd=0s?

  1. In Fig. 8, the quadratic adaption has less median deviations than the linear adaption when Δtd<3s. What condition is it appropriate to use linear adaption? If Δtd is an key factor, whether this parameter can be selected automatically in inspecting the wrought-hot specimen?

Author Response

Fig. 2(b) shows the decrease of pixel intensity and the infrared emission intensity during cooling period. Line 136-138 indicates that the variation is in a quadratic form for 1.3s~3s and a linear form for 3s~6s. However, the infrared emission intensity is influenced by the emissive coefficient of the object surface and the type of infrared cameras. What’s the range of wavelength of the infrared camera?
I have added the range of the used infrared camera in Sec. 2.5.

If different materials and surface emission coefficients are used for the inspection, this curve may be varied.
The used measurement object is identical for each experiment. Additionally, Fig. 2 (b) shows the emitted thermal intensity in kW/m2, which is the direct measurement value of an infrared camera.  I have added clarifying remarks in the caption of Fig. 2 and in the text.

What’s the distance from the 3d measurement setup to the detected object? Does the change of distance influence the illumination and light deflection?
I have added the distance from the optical center of the projector to the measurement object in Sec. 2.5. The distance between the measurement system and the measurement object as well as the angle between the refractive index field and each light ray in influencing the light deflection. An exhaustive investigation of this effect was conducted in [17, 19] and is well beyond the scope of this paper and therefore omitted.   

Many symbols are undefined, e.g., line 154, Δia,nom and Δia,den; line 176, Iest(u); in Fig. 3, nm; in Fig. 4(a), Ip; in Eq. 17, Nm.
I have added the definitions or modified the necessary equations with the exception of Ip, which was defined in the context of Eq. (1).

Be careful about writing, e.g. line 170, “is is”; line 307, “TThe”.
Thank you for the hint. I performed an extensive spelling and grammar check. 

The linear and quadratic compensation is given in Fig. 4(a). However, the mathematical expression of these two functions should be presented in the paper.
I have added the equations of the influences as Eqs. (16,17).

In line 211, why is the intensity gradient set to 2 per image in the experiment? Discussion is needed since this is important for the precision 3d measurement.
This value is set corresponding to the linear influence of the simulation model. I have added a corresponding reference in the text in Sec. 3.3.

In Fig. 8, please explain why the measurement without the flow actuator using a regular 8-image sequence achieves the smallest median deviation when Δtd=0s? Does it show that the proposed method is not applicable when Δtd=0s?
I myself do not know why the measurement without the flow actuator and the regular 8-image sequence shows the smallest median deviation for Δtd=0s. The previous results, e.g., shown in [17], indicate that the established reconstruction quality metric might not be perfect to assess the magnitude of the light deflection effect, since additional effects are also considered like, e.g., the intensity transient described in this paper. However, the results from the background-oriented schlieren measurements from [17] also show the presence of a light deflecting field around a hot specimen as long as the flow actuator is deactivated. Condensing all the given information, the reconstruction quality metric might have failed to generate valid results for Δtd=0s, the linear approximation and the deactivated flow actuator. I enhanced the discussion section accordingly.

In Fig. 8, the quadratic adaption has less median deviations than the linear adaption when Δtd<3s. What condition is it appropriate to use linear adaption? If Δtd is a key factor, whether this parameter can be selected automatically in inspecting the wrought-hot specimen?
This is a very good question I have not thought about before. Δtd is a key factor since it is one of the influences on the form and magnitude of the intensity transient. Other influences include the heat capacity of the object combined with its temperature as well as the ‘strength’ of the forced air flow. Since the air flow actuator needs to be adapted to each new measurement object (cf. [17]), there probably is not an easy to formulate algorithm for this. My best guess is that a semi-automatic in-situ guess of Δtd could be based on an elongated projection of the high-frequency patterns in combination with Eq. (19) or Eq. (21). An investigation of this matter is highly interesting but would also be beyond the scope of this paper. I have added the ‘Future work’ section accordingly.

Reviewer 2 Report

General remarks

In this paper, the authors develop a theoretical approach to compensate for the inhomogeneous refractive index of the hot air surrounding a hot object during measurement by fringe projection profilometry. The paper has great academic merit and provides an unconventional approach to improving accuracy in non-contact optical measurements of wrought-hot specimens. However, my main remark is that the contributions are not sufficiently well placed in a more general context of difficult-to-measure surfaces [1,2]. How do these results generalize for other similar applications? What is the main takeaway? When is it best to use the proposed method? Furthermore, the title does not reflect well the contents of the paper; it is too ambiguous. 

Although the writing needs to be polished and the article is often over verbose, each section explains with sufficient detail the considered topic. However, the paper follows an unconventional structure. There is a discussion and conclusions section in the middle of the paper. To improve readability, the introduction section should describe better how the paper is organized. Also, the discussion and summary sections are too long and repeat what was already described. I urge the authors to reconsider shortening conclusions to provide the key message they wish to convey concisely. Consider an interested reader that skims through the paper. Try to deliver your conclusion in a broader context of 3D surface profilometry of challenging surfaces. 

Minor remarks

- The abstract could be a bit more specific. 
- Abstract typos. “measurement”, “influenced”
- Line 40. “ previously established reconstruction quality metric.” previously established where or by whom?
- Line 66. “Peng” missing citation.
- Line 170. “is is”
- Try to avoid writing sentences with an unclear antecedent, like lines 196 and 200. It may be unclear who or what “This” refers to. There are other sentences with this problem.
- Line 215. “For a better comparability”, try to rewrite for clarity. 
- Line 216. “exclusively”.
- Line 334. “conducted” -> concluded?
- Line 408. “outlast the deficits” -> outweigh the drawbacks?

[1] Landmann, Martin, et al. "3D shape measurement of objects with uncooperative surface by projection of aperiodic thermal patterns in simulation and experiment." Optical Engineering 59.9 (2020): 094107.
[2]    A. G. Marrugo, F. Gao, and S. Zhang, “State-of-the-art active optical techniques for three-dimensional surface metrology: a review [Invited],” J Opt Soc Am A Opt Image Sci Vis, vol. 37, no. 9, pp. B60–18, 2020.

Author Response

In this paper, the authors develop a theoretical approach to compensate for the inhomogeneous refractive index of the hot air surrounding a hot object during measurement by fringe projection profilometry. The paper has great academic merit and provides an unconventional approach to improving accuracy in non-contact optical measurements of wrought-hot specimens.
Thank you for your kind assessment of my work.

However, my main remark is that the contributions are not sufficiently well placed in a more general context of difficult-to-measure surfaces [1,2].
I agree with your critique on the missing placement within the context of difficult-to-measure surfaces. However, in my opinion my paper does not belong into this concept. A wrought-hot red-glowing surface is per se not difficult to measure (unlike, e.g., translucent, transparent or black surfaces), it just introduces a different kind of challenge to the measurement. Nevertheless, I have added your proposed references to the introduction since [2] gives an nice and thorough review to optical 3d measurement and [1] is a good example of difficult to measure surfaces.

How do these results generalize for other similar applications? What is the main takeaway?
I have added the main takeaway to the conclusion. The sequence including the changed order of the projection images may be used in any given case, since it does no induce additional costs but reduces the influence of possible intensity transients. The use of the enhanced adapted sequence is limited to deterministic intensity transients, such as Newtonian cooling.

When is it best to use the proposed method?
I have tried to include separate use cases for the proposed method in the ‘Conclusions’ section. 

Furthermore, the title does not reflect well the contents of the paper; it is too ambiguous. 
I changed the title and included an example.

Although the writing needs to be polished and the article is often over verbose, each section explains with sufficient detail the considered topic. However, the paper follows an unconventional structure. There is a discussion and conclusions section in the middle of the paper. To improve readability, the introduction section should describe better how the paper is organized.
I have improved the introduction accordingly. It now contains a

Also, the discussion and summary sections are too long and repeat what was already described. I urge the authors to reconsider shortening conclusions to provide the key message they wish to convey concisely. Consider an interested reader that skims through the paper. Try to deliver your conclusion in a broader context of 3D surface profilometry of challenging surfaces. 
I have structured the former ‘Summary, conclusions and Future Work’ section more strictly into their own sections. This enables a reader skimming through an easier access to the desired information.

Minor remarks:
- The abstract could be a bit more specific. 
I have specified the effect leading to the described influence of the intensity transient.

- Abstract typos. “measurement”, “influenced”
Fixed.

- Line 40. “previously established reconstruction quality metric.” previously established where or by whom?
I have shown the source for on the ‘previously established reconstruction metric as my own paper in l.20.

- Line 66. “Peng” missing citation.
Included citation [23].

- Line 170. “is is”
Fixed.
- Try to avoid writing sentences with an unclear antecedent, like lines 196 and 200. It may be unclear who or what “This” refers to. There are other sentences with this problem.
I tried to identify said sentences and fixed those.

- Line 215. “For a better comparability”, try to rewrite for clarity. 
Fixed.

- Line 216. “exclusively”.
Fixed.

- Line 334. “conducted” -> concluded?
Fixed.

- Line 408. “outlast the deficits” -> outweigh the drawbacks?
Yes.

Reviewer 3 Report

The Paper is overall well written, but needs some clarification / correction:

Section 2.1: clarify variable u as a vector with its entries
Eq (4) Why is f base ^0 =1 and not set to some f base value but always to 1

line 69: one . too much

line 77: transition from I a to Ia k unclear. How do you get from Eq. (8) to (9) to (10)?

line 78: why from 4k to 2(k-1)+4 and not the other way around?

Figures, generally: Place figures below introduction in text.

Figure captions: Explanations too scarce, write what is to be seen and what it means! Fig. 1 too scarce,

Fig. 2: a: scale missing – size of the object unknown, b: … the data – what data? Which data to which axis? What is the interpretation?

Figure 1: Add schematic drawing, picture is difficult for showing the geometry

Eq.13: avoid i as it stands for SQRT(-1)

Table 1 / Fig. 3 confusing: Place Tab. 1 below Line 169 and Fig. 3 below line 176

Line 204: symbol %o

Figure 4 below line 204, add interpretation of data in Fig. caption

Line 224: avoid * as it stand for complex conjugate

Line 249 why points with 2 and 4 parameters?

Line 288. Definition of terms?

Figure 8: Confusing! Add explanation to fig. caption: What do we learn from the diagram? (In principle the information is somewhere) Too many colors and symbols.

Important: Fig. 8 only shows the median deviation – from what? -how defined? No absolute scale of the measurement /object is given. State absolute size. Are these relatively small or large errors?

Author Response

Section 2.1: clarify variable u as a vector with its entries
I have added the definition.

Eq (4) Why is f base ^0 =1 and not set to some f base value but always to 1
The used multi-frequency phase-shift unwrapping algorithm needs a frequency of  to successfully perform the unwrapping.  

line 69: one . too much
Fixed.

line 77: transition from I a to Ia k unclear. How do you get from Eq. (8) to (9) to (10)?
I have tried to clarify the confusion by using different formula symbols. In this part of the paper, I tried to differentiate between the general form as   and the expression in relation to a frequency coefficient  .

line 78: why from 4k to 2(k-1)+4 and not the other way around?
This is from the example in [25]. The original algorithm uses , i.e., three times four projection images (). The sequence from [25] uses two projections of  and  and four projections of , so  images.

Figures, generally: Place figures below introduction in text.
I let latex place the figures itself. If the paper is accepted, I will check the placement in the page proof.

Figure captions: Explanations too scarce, write what is to be seen and what it means! Fig. 1 too scarce,
An exhaustive description of the optical setup was conducted in [17] and of the flow actuator setup in [19]. Repeating these descriptions is beyond the scope of this paper and was therefore omitted. I have added this reference in the corresponding text body.

 Fig. 2: a: scale missing – size of the object unknown, b: … the data – what data? Which data to which axis? What is the interpretation?
A description of the used measurement object and the interpretation of the data was conducted in the text and was therefore omitted in the caption. I have added the axis to which the data points are related in the caption.

Figure 1: Add schematic drawing, picture is difficult for showing the geometry.
See above.

Eq.13: avoid i as it stands for SQRT(-1)
IMHO  represents the index of an image and is always subject to definition.

Table 1 / Fig. 3 confusing: Place Tab. 1 below Line 169 and Fig. 3 below line 176
As for the other image, I will check the placement in the page proof if the paper is accepted.

Line 204: symbol %o
I like the written ‘per mille’ in the given context better.

Figure 4 below line 204, add interpretation of data in Fig. caption
See above. I do not like to add the interpretation to a figure caption, as it often enlarges the caption excessively.

Line 224: avoid * as it stand for complex conjugate
See above. I like defining the mathematical parameters in the text, e.g., as I like to use ‘’ to indicate a vector product.

Line 249 why points with 2 and 4 parameters?
These points correspond to Eqs. (8)-(10), where a different number of parameters is used. I have added the explicit reference in the text.

Line 288. Definition of terms?
I have added the corresponding definitions.

Figure 8: Confusing! Add explanation to fig. caption: What do we learn from the diagram? (In principle the information is somewhere) Too many colors and symbols.
I have added to the description and explanation to the text.

Important: Fig. 8 only shows the median deviation – from what? -how defined? No absolute scale of the measurement /object is given. State absolute size. Are these relatively small or large errors?
I have added clarifying remarks regarding which the corresponding reconstructed value. The results are not subject to some unknown influences, so its interpretation is not as clear as I would like it to be. I have enhanced the discussion on this topic to clarify the made assumptions and consider the drawbacks of the used metric.

Round 2

Reviewer 1 Report

The authors well addressed concerns raised bt the reviewer. The quality of the manuscript is in good shape for publication.